# Simple Policy Optimization

**Zhengpeng Xie** [* 1]   **Qiang Zhang** [* 1 2]   **Fan Yang** [* 3]   **Marco Hutter** [3]   **Renjing Xu** [1]

## Abstract

Model-free reinforcement learning algorithms have seen remarkable progress, but key challenges remain. Trust Region Policy Optimization (TRPO) is known for ensuring monotonic policy improvement through conservative updates within a trust region, backed by strong theoretical guarantees. However, its reliance on complex second-order optimization limits its practical efficiency. Proximal Policy Optimization (PPO) addresses this by simplifying TRPO's approach using ratio clipping, improving efficiency but sacrificing some theoretical robustness. This raises a natural question: Can we combine the strengths of both methods? In this paper, we introduce *Simple Policy Optimization* (SPO), a novel unconstrained first-order algorithm. By slightly modifying the policy loss used in PPO, SPO can achieve the best of both worlds. Our new objective improves upon ratio clipping, offering stronger theoretical properties and better constraining the probability ratio within the trust region. Empirical results demonstrate that SPO outperforms PPO with a simple implementation, particularly for training large, complex network architectures end-to-end.

**Code is available at** *Simple-Policy-Optimization*.

## 1. Introduction

Deep Reinforcement Learning (DRL) has achieved great success in recent years, notably in games (Mnih et al., 2015; Silver et al., 2016; 2017; 2018; Vinyals et al., 2019), foundation model fine-tuning (Ouyang et al., 2022; Black et al., 2023), and robotic control (Makoviychuk et al., 2021; Rudin et al., 2022). Policy gradient (PG) methods (Sutton & Barto, 2018; Lehmann, 2024), as a major paradigm in RL, have

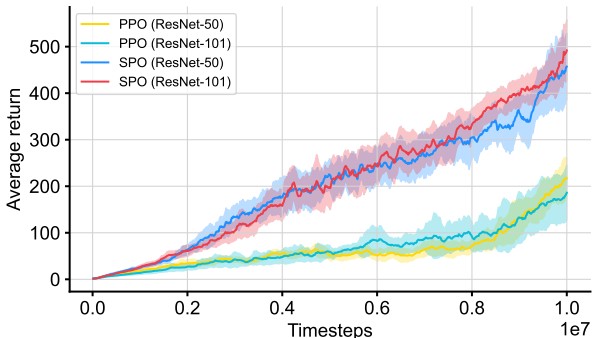

**Figure 1.** Training performance in the *Breakout* environment. SPO is a novel model-free algorithm capable of end-to-end training for extremely deep neural network architectures, positioning itself as a promising alternative to the well-known PPO algorithm.

been widely adopted by the academic community. One main practical challenge of PG methods is to reduce the variance of the gradients while keeping the bias low (Sutton et al., 2000; Schulman et al., 2015b). In this context, a widely used technique is to add a baseline when sampling an estimate of the action-value function (Greensmith et al., 2004). Another challenge of PG methods is to estimate the proper step size for the policy update (Kakade & Langford, 2002; Schulman et al., 2015a). Given that the training data strongly depends on the current policy, a large step size may result in a collapse of policy performance, whereas a small one may impair the sample efficiency of the algorithm.

To address these challenges, Schulman et al. (2015a) proved that optimizing a certain surrogate objective guarantees policy improvement with non-trivial step sizes. Subsequently, the TRPO algorithm was derived through a series of approximations, which impose a trust region constraint during the policy iterations, leading to monotonic policy improvement in theory. However, given the complexity of second-order optimization, TRPO is highly inefficient and can be hard to extend to large-scale RL environments. Proximal Policy Optimization (PPO) (Schulman et al., 2017) is designed to enforce comparable constraints on the difference between successive policies during the training process, while only using first-order optimization. By clipping the current data that exceeds the probability ratio limit to a constant, PPO attempts to remove the high incentive for pushing the cur-

*Equal contribution [1]The Hong Kong University of Science and Technology (Guangzhou) [2]Beijing Innovation Center of Humanoid Robotics [3]ETH Zurich. Correspondence to: Zhengpeng Xie <zhengpengxie@hkust-gz.edu.cn>.

*Proceedings of the 42nd International Conference on Machine Learning*, Vancouver, Canada. PMLR 267, 2025. Copyright 2025 by the author(s).

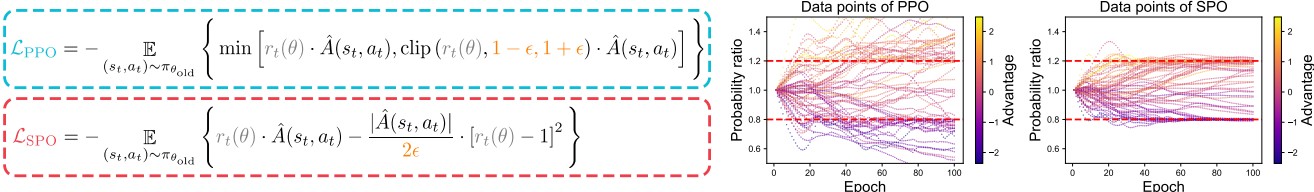

$$\mathcal{L}_{\mathrm{PPO}} = - \mathop{\mathbb{E}}_{(s_t,a_t) \sim \pi_{\theta_{\mathrm{old}}}} \left\{ \min \left[ r_t(\theta) \cdot \hat{A}(s_t, a_t), \mathrm{clip}\left( r_t(\theta), 1-\epsilon, 1+\epsilon \right) \cdot \hat{A}(s_t, a_t) \right] \right\}$$

$$\mathcal{L}_{\mathrm{SPO}} = - \mathop{\mathbb{E}}_{(s_t,a_t) \sim \pi_{\theta_{\mathrm{old}}}} \left\{ r_t(\theta) \cdot \hat{A}(s_t, a_t) - \frac{|\hat{A}(s_t, a_t)|}{2\epsilon} \cdot [r_t(\theta) - 1]^2 \right\}$$

*Figure 2.* (Left) The only difference between SPO and PPO is the policy loss, where $r_t(\theta) = \pi_\theta(a_t|s_t)/\pi_{\theta_{\mathrm{old}}}(a_t|s_t)$ and $\epsilon$ is the probability ratio hyperparameter, making it simple and straightforward to implement SPO based on high-quality PPO implementations. (Right) The optimization behavior of PPO and SPO is visualized, where each scatter point represents the probability ratio of a single data point for a specific training epoch, with its color corresponding to its advantage, and the red line representing the probability ratio bound.

rent policy away from the old one. It has been demonstrated that PPO can be effectively extended to large-scale complex control tasks (Ye et al., 2020; Makoviychuk et al., 2021).

Despite its success, the optimization behavior of PPO remains insufficiently understood. Although PPO aims to constrain the probability ratio deviations between successive policies, it often fails to keep these ratios within bounds (Ilyas et al., 2018; Engstrom et al., 2020; Wang et al., 2020). In some tasks, the ratios can even escalate to values as high as 40 (Wang et al., 2020). Furthermore, studies have revealed that PPO's performance is highly dependent on "code-level optimizations" (Andrychowicz et al., 2021; Huang et al., 2022a). The implementation of PPO includes numerous code-level details that critically influence its effectiveness (Engstrom et al., 2020; Huang et al., 2022b).

In this paper, we propose a new model-free RL algorithm named Simple Policy Optimization (SPO) designed to more effectively bound probability ratios through a novel objective function. The key differences in optimization behavior between PPO and SPO are illustrated in Figure 2. Our main contributions are summarized as follows:

- We theoretically prove that optimizing a tighter performance lower bound using Total Variation (TV) divergence constrained space results in more consistent policy improvement.

- To overcome PPO's limitation in constraining probability ratios, we propose a new objective function, leading to the development of the proposed SPO algorithm.

- Experiments benchmark various policy gradient algorithms across different environments, showing that SPO can achieve competitive performance with a simple implementation, improved sample efficiency, and easier training of deeper policy networks.

## 2. Related Work

Since TRPO (Schulman et al., 2015a) theoretically demonstrated monotonic policy improvement, numerous studies

have explored how to enforce trust region constraints efficiently, which are essential for ensuring robust policy improvement. For instance, the widely-used PPO algorithm (Schulman et al., 2017) was the first to introduce the heuristic clipping technique, effectively avoiding the computationally expensive second-order optimization. This heuristic clipping technique has been widely used in various reinforcement learning algorithms (Queeney et al., 2021; Zhuang et al., 2023; Gan et al., 2024).

However, empirical evidence from a wide range of studies demonstrates that ratio clipping fails to enforce trust region constraints effectively (Wang et al., 2020). To prevent aggressive policy updates, previous works have focused on designing adaptive learning rates based on TV divergence or KL divergence (Heess et al., 2017; Queeney et al., 2021; Rudin et al., 2022), which have been shown to effectively enhance the stability of PPO. On the other hand, code-level optimizations are crucial for the robust performance of PPO (Engstrom et al., 2020). High-quality implementations of PPO involve numerous code details (Huang et al., 2022a;b), making it challenging to accurately assess the core factors that truly affect the algorithm's performance.

In this work, we argue that heuristic clipping technique cannot enforce trust region constraints (see Figure 2). During PPO's iterations, ratio clipping zeros the gradients of certain data points, which can lead to a lack of corrective gradients to prevent the policy from escaping the trust region, thus undermining the monotonic improvement guarantee. As a result, PPO requires additional code-level tuning, such as adaptive learning rates or early stopping strategies, to artificially prevent performance collapse. We reveal this inherent flaw of ratio clipping and propose promising alternatives.

## 3. Background

### 3.1. Reinforcement Learning

Online reinforcement learning is a mathematical framework for sequential decision-making, which is generally defined by the Markov Decision Process (MDP) $\mathcal{M} = (\mathcal{S}, \mathcal{A}, r, \mathcal{P}, \rho_0, \gamma)$, where $\mathcal{S}$ and $\mathcal{A}$ represent the state space

and action space, $r : \mathcal{S} \times \mathcal{A} \mapsto \mathbb{R}$ is the reward function, $\mathcal{P} : \mathcal{S} \times \mathcal{A} \times \mathcal{S} \mapsto [0,1]$ is the probability distribution of the state transition function, $\rho_0 : \mathcal{S} \mapsto [0,1]$ is the initial state distribution, while $\gamma \in (0,1)$ is the discount factor.

Suppose that an agent interacts with the environment following policy $\pi$, i.e., $a_t \sim \pi(\cdot|s_t)$ and obtains a trajectory $\tau = (s_0, a_0, r_0, \ldots, s_t, a_t, r_t, \ldots)$, where $r_t = r(s_t, a_t)$. The goal of RL is to learn a policy that maximizes the objective $\eta(\pi) = \mathbb{E}_{\tau \sim \pi} [\sum_{t=0}^{\infty} \gamma^t r_t]$, where the notation $\mathbb{E}_{\tau \sim \pi}$ represents the expected return of the trajectory $\tau$ generated by the agent following policy $\pi$, i.e., $s_0 \sim \rho_0(\cdot), a_t \sim \pi(\cdot|s_t), r_t = r(s_t, a_t), s_{t+1} \sim \mathcal{P}(\cdot|s_t, a_t)$. The action-value function and value function are defined as

$$Q_\pi(s_t, a_t) = \mathbb{E}_{s_{t+1}, a_{t+1}, \ldots} \left[ \sum_{k=0}^{\infty} \gamma^k r(s_{t+k}, a_{t+k}) \right], \quad (1)$$

$$V_\pi(s_t) = \mathbb{E}_{a_t \sim \pi(\cdot|s_t)} [Q_\pi(s_t, a_t)].$$

Given $Q_\pi$ and $V_\pi$, the advantage function can be expressed as $A_\pi(s_t, a_t) = Q_\pi(s_t, a_t) - V_\pi(s_t)$.

## 3.2. Trust Region Policy Optimization

Classic policy gradient methods cannot reuse data and are highly sensitive to the hyperparameters. To address these issues, in Trust Region Policy Optimization (TRPO), Schulman et al. (2015a) derived a lower bound for policy improvement. Before that, Kakade & Langford (2002) first proved the following policy performance difference theorem.

**Theorem 3.1.** *(Kakade & Langford, 2002) Let $\mathbb{P}(s_t = s|\pi)$ represents the probability of the $t$-th state equals to $s$ in trajectories generated by the agent following policy $\pi$, and $\rho_\pi(s) = (1 - \gamma) \sum_{t=0}^{\infty} \gamma^t \mathbb{P}(s_t = s|\pi)$ represents the normalized discounted visitation distribution. Given any two policies, $\pi$ and $\tilde{\pi}$, their performance difference can be measured by*

$$\eta(\tilde{\pi}) - \eta(\pi) = \frac{1}{1 - \gamma} \mathbb{E}_{s \sim \rho_{\tilde{\pi}}(\cdot), a \sim \tilde{\pi}(\cdot|s)} [A_\pi(s, a)]$$

$$= \frac{1}{1 - \gamma} \sum_s \rho_{\tilde{\pi}}(s) \sum_a \tilde{\pi}(a|s) \cdot A_\pi(s, a), \quad (2)$$

*where $\eta(\pi) = \mathbb{E}_{\tau \sim \pi} [\sum_{t=0}^{\infty} \gamma^t r_t]$.*

The key insight is that the new policy $\tilde{\pi}$ will improve (or at least remain constant) as long as it has a nonnegative expected advantage at every state $s$. Then, the following performance improvement lower bound is given:

**Theorem 3.2.** *(Achiam et al., 2017) Given any two policies, $\pi$ and $\tilde{\pi}$, the following bound holds:*

$$\eta(\tilde{\pi}) - \eta(\pi) \geq \frac{1}{1 - \gamma} \mathbb{E}_{s \sim \rho_\pi(\cdot), a \sim \tilde{\pi}(\cdot|s)} [A_\pi(s, a)]$$

$$- \frac{2\xi\gamma}{(1 - \gamma)^2} \mathbb{E}_{s \sim \rho_\pi(\cdot)} [D_{\text{TV}}(\pi \| \tilde{\pi})[s]], \quad (3)$$

*where $\xi = \max_s |\mathbb{E}_{a \sim \tilde{\pi}(\cdot|s)} [A_\pi(s, a)]|$, $D_{\text{TV}}$ is the Total Variation (TV) divergence.*

Using importance sampling on action $a \sim \tilde{\pi}(\cdot|s)$ and according to the Pinsker's inequality, we have

$$\eta(\tilde{\pi}) - \eta(\pi) \geq \frac{1}{1 - \gamma} \mathbb{E}_{s \sim \rho_\pi(\cdot), a \sim \pi(\cdot|s)} \left[ \frac{\tilde{\pi}(a|s)}{\pi(a|s)} \cdot A_\pi(s, a) \right]$$

$$- \frac{2\xi\gamma}{(1 - \gamma)^2} \mathbb{E}_{s \sim \rho_\pi(\cdot)} \left[ \sqrt{\frac{1}{2} D_{\text{KL}}(\pi \| \tilde{\pi})[s]} \right]. \quad (4)$$

At this point, the subscripts of the expectation in (2) are replaced from $s \sim \rho_{\tilde{\pi}}(\cdot)$ and $a \sim \tilde{\pi}(\cdot|s)$ to $s \sim \rho_\pi(\cdot)$ and $a \sim \pi(\cdot|s)$, which means that we can now reuse the current data. In TRPO, the lower bound in (4) is *indirectly* optimized by solving the following optimization problem:

$$\max_\theta \ \mathbb{E}_{(s_t, a_t) \sim \pi_{\theta_{\text{old}}}} \left[ \frac{\pi_\theta(a_t|s_t)}{\pi_{\theta_{\text{old}}}(a_t|s_t)} \cdot \hat{A}(s_t, a_t) \right], \quad (5)$$

$$\text{s.t.} \ \mathbb{E} [D_{\text{KL}}(\pi_{\theta_{\text{old}}} \| \pi_\theta)] \leq \delta.$$

This problem includes a constraint where $\delta$ is a hyperparameter that limits the KL divergence between successive policies, with $\hat{A}(s_t, a_t)$ being the estimate of the advantage function, and the objective is called "surrogate objective".

## 3.3. Proximal Policy Optimization

Due to the necessity of solving a constrained optimization problem (5) in each update, TRPO is highly inefficient and can be challenging to apply to large-scale reinforcement learning tasks.

Schulman et al. (2017) proposed a new objective called "clipped surrogate objective", in which the algorithm is named Proximal Policy Optimization (PPO). PPO retains similar constraints of TRPO but is much easier to implement and involves only first-order optimization.

The "clipped surrogate objective", also called PPO-Clip, adopts a ratio clipping function. Denote $\hat{A}_t = \hat{A}(s_t, a_t)$, the objective of PPO-Clip can be expressed as

$$J_{\text{clip}}(\theta) = \mathbb{E}_{(s_t, a_t) \sim \pi_{\theta_{\text{old}}}} \left\{ \min \left[ r_t(\theta) \cdot \hat{A}_t, \tilde{r}_t(\theta) \cdot \hat{A}_t \right] \right\}, \quad (6)$$

where

$$r_t(\theta) = \frac{\pi_\theta(a_t|s_t)}{\pi_{\theta_{\text{old}}}(a_t|s_t)}, \quad \tilde{r}_t(\theta) = \text{clip}(r_t(\theta), 1 - \epsilon, 1 + \epsilon), \quad (7)$$

with $\pi_{\theta_{\text{old}}}$ and $\pi_\theta$ being the old policy and the current policy. The gradient of PPO-Clip, given the training data $(s_t, a_t)$,

can be expressed as

$$\nabla_\theta J_{\mathrm{clip}}(\theta) = \begin{cases} \nabla_\theta r_t(\theta) \cdot \hat{A}_t, & \hat{A}_t > 0, r_t(\theta) \leq 1 + \epsilon; \\ \nabla_\theta r_t(\theta) \cdot \hat{A}_t, & \hat{A}_t < 0, r_t(\theta) \geq 1 - \epsilon; \\ 0, & \text{otherwise.} \end{cases}$$

(8)

In other words, PPO-Clip aims to remove the high incentive for pushing the current policy away from the old one. PPO-Clip has gained wide adoption in the academic community due to its simplicity and performance.

## 4. Methodology

PPO attempts to limit the differences between successive policies through ratio clipping. However, Wang et al. (2020) proved the following theorem:

**Theorem 4.1.** *(Wang et al., 2020) For discrete action space tasks where $|\mathcal{A}| \geq 3$ or continuous action space tasks where the output of the policy $\pi_\theta$ follows a multivariate Gaussian distribution. Let $\Theta = \{\theta | 1 - \epsilon \leq r_t(\theta) \leq 1 + \epsilon\}$, we have $\sup_{\theta \in \Theta} D_{\mathrm{KL}}(\pi_{\theta_{\mathrm{old}}} \| \pi_\theta)[s_t] = +\infty$ for both discrete and continuous action space tasks.*

Theorem 4.1 demonstrates that $D_{\mathrm{KL}}(\pi_{\theta_{\mathrm{old}}} \| \pi_\theta)[s_t]$ is not necessarily bounded even if the probability ratio $r_t(\theta)$ is bounded. However, this theorem considers only an extreme case involving a single data point, which is less typical than the batch processing used in training data. On a broader scale, the heuristic clipping technique employed by PPO aims to bound the TV divergence for sufficient batch sizes (Queeney et al., 2021). This relationship is formalized as

$$\mathbb{E}_{s \sim \rho_\pi(\cdot)} [D_{\mathrm{TV}}(\pi \| \tilde{\pi})[s]] = \frac{1}{2} \mathop{\mathbb{E}}_{\substack{s \sim \rho_\pi(\cdot) \\ a \sim \pi(\cdot|s)}} \left[ \left| \frac{\tilde{\pi}(a|s)}{\pi(a|s)} - 1 \right| \right].$$

(9)

Then, the performance improvement lower bound (3) can be rewritten as

$$\eta(\tilde{\pi}) - \eta(\pi) \geq \frac{1}{1 - \gamma} \mathbb{E}_{s \sim \rho_\pi(\cdot), a \sim \pi(\cdot|s)} \left[ \frac{\tilde{\pi}(a|s)}{\pi(a|s)} \cdot A_\pi(s, a) \right] - \frac{\xi \gamma}{(1 - \gamma)^2} \mathbb{E}_{s \sim \rho_\pi(\cdot), a \sim \pi(\cdot|s)} \left[ \left| \frac{\tilde{\pi}(a|s)}{\pi(a|s)} - 1 \right| \right].$$

(10)

This explains why PPO attempts to limit the probability ratio $|\tilde{\pi}(a|s)/\pi(a|s) - 1| \leq \epsilon$, as this enforces a TV divergence trust region in expectation.

Finally, we also found that PPO, which aims to bound the TV divergence, can offer a larger solution space compared to methods that incorporate a looser KL divergence as a constraint (e.g., in TRPO). To illustrate this, we present the following proposition:

**Proposition 4.2.** *Given the old policy $\pi$, define the solution spaces under the TV and KL divergence constraints as*

*follows:*

$$\begin{aligned} \Omega_{\mathrm{TV}} &= \{\tilde{\pi} \mid D_{\mathrm{TV}}(\pi \| \tilde{\pi})[s] \leq \delta_{\mathrm{TV}}, \forall s \in \mathcal{S}\}, \\ \Omega_{\mathrm{KL}} &= \{\tilde{\pi} \mid D_{\mathrm{KL}}(\pi \| \tilde{\pi})[s] \leq \delta_{\mathrm{KL}}, \forall s \in \mathcal{S}\}, \end{aligned}$$

(11)

*where $\delta_{\mathrm{KL}} > 0$ is a predefined threshold. Let $\delta_{\mathrm{TV}} \geq \sqrt{\frac{1}{2}\delta_{\mathrm{KL}}}$, we establish that $\Omega_{\mathrm{KL}} \subset \Omega_{\mathrm{TV}}$.*

*Proof.* For any given $\delta_{\mathrm{KL}}$ and $\tilde{\pi} \in \Omega_{\mathrm{KL}}$, using Pinsker's inequality, we have $D_{\mathrm{TV}}(\pi \| \tilde{\pi})[s] \leq \sqrt{\frac{1}{2}D_{\mathrm{KL}}(\pi \| \tilde{\pi})[s]} \leq \sqrt{\frac{1}{2}\delta_{\mathrm{KL}}} \leq \delta_{\mathrm{TV}}$, therefore $\tilde{\pi} \in \Omega_{\mathrm{KL}} \implies \tilde{\pi} \in \Omega_{\mathrm{TV}}$, which means $\Omega_{\mathrm{KL}} \subset \Omega_{\mathrm{TV}}$, concluding the proof. $\square$

Additionally, the optimal solution to the lower bound in the TV divergence solution space, $\Omega_{\mathrm{TV}}$, is expected to be superior. We now present the following theorem:

**Theorem 4.3.** *Given the old policy $\pi$, and $\Omega_{\mathrm{TV}}, \Omega_{\mathrm{KL}}$ presented in Proposition 4.2, let*

$$\begin{aligned} \mathcal{L}_\pi^{\mathrm{TV}}(\tilde{\pi}) =& \frac{1}{1 - \gamma} \mathbb{E}_{s \sim \rho_\pi(\cdot), a \sim \pi(\cdot|s)} \left[ \frac{\tilde{\pi}(a|s)}{\pi(a|s)} \cdot A_\pi(s, a) \right] \\ &- \frac{2\xi\gamma}{(1 - \gamma)^2} \mathbb{E}_{s \sim \rho_\pi(\cdot)} [D_{\mathrm{TV}}(\pi \| \tilde{\pi})[s]], \\ \mathcal{L}_\pi^{\mathrm{KL}}(\tilde{\pi}) =& \frac{1}{1 - \gamma} \mathbb{E}_{s \sim \rho_\pi(\cdot), a \sim \pi(\cdot|s)} \left[ \frac{\tilde{\pi}(a|s)}{\pi(a|s)} \cdot A_\pi(s, a) \right] \\ &- \frac{2\xi\gamma}{(1 - \gamma)^2} \mathbb{E}_{s \sim \rho_\pi(\cdot)} \left[ \sqrt{\frac{1}{2}D_{\mathrm{KL}}(\pi \| \tilde{\pi})[s]} \right]. \end{aligned}$$

(12)

*be the lower bounds of performance improvement with TV divergence and KL divergence. Let $\delta_{\mathrm{TV}} \geq \sqrt{\frac{1}{2}\delta_{\mathrm{KL}}}$, denote*

$$\tilde{\pi}_{\mathrm{TV}}^* = \arg\max_{\tilde{\pi} \in \Omega_{\mathrm{TV}}} \mathcal{L}_\pi^{\mathrm{TV}}(\tilde{\pi}), \quad \tilde{\pi}_{\mathrm{KL}}^* = \arg\max_{\tilde{\pi} \in \Omega_{\mathrm{KL}}} \mathcal{L}_\pi^{\mathrm{KL}}(\tilde{\pi}),$$

(13)

*then $\mathcal{L}_\pi^{\mathrm{TV}}(\tilde{\pi}_{\mathrm{TV}}^*) \geq \mathcal{L}_\pi^{\mathrm{KL}}(\tilde{\pi}_{\mathrm{KL}}^*)$.*

*Proof.* Since $\Omega_{\mathrm{KL}} \subset \Omega_{\mathrm{TV}}$, we have

$$\begin{aligned} &\mathcal{L}_\pi^{\mathrm{TV}}(\tilde{\pi}_{\mathrm{TV}}^*) \geq \mathcal{L}_\pi^{\mathrm{TV}}(\tilde{\pi}_{\mathrm{KL}}^*) = \\ &\frac{1}{1 - \gamma} \mathbb{E}_{s \sim \rho_\pi(\cdot), a \sim \pi(\cdot|s)} \left[ \frac{\tilde{\pi}_{\mathrm{KL}}^*(a|s)}{\pi(a|s)} \cdot A_\pi(s, a) \right] \\ &- \frac{2\xi\gamma}{(1 - \gamma)^2} \mathbb{E}_{s \sim \rho_\pi(\cdot)} [D_{\mathrm{TV}}(\pi \| \tilde{\pi}_{\mathrm{KL}}^*)[s]] \geq \\ &\frac{1}{1 - \gamma} \mathbb{E}_{s \sim \rho_\pi(\cdot), a \sim \pi(\cdot|s)} \left[ \frac{\tilde{\pi}_{\mathrm{KL}}^*(a|s)}{\pi(a|s)} \cdot A_\pi(s, a) \right] \\ &- \frac{2\xi\gamma}{(1 - \gamma)^2} \mathbb{E}_{s \sim \rho_\pi(\cdot)} \left[ \sqrt{\frac{1}{2}D_{\mathrm{KL}}(\pi \| \tilde{\pi}_{\mathrm{KL}}^*)[s]} \right] \\ &= \mathcal{L}_\pi^{\mathrm{KL}}(\tilde{\pi}_{\mathrm{KL}}^*), \end{aligned}$$

(14)

thus $\mathcal{L}_\pi^{\mathrm{TV}}(\tilde{\pi}_{\mathrm{TV}}^*) \geq \mathcal{L}_\pi^{\mathrm{KL}}(\tilde{\pi}_{\mathrm{KL}}^*)$, concluding the proof. $\square$

---

**Algorithm 1** Simple Policy Optimization (SPO)

1: **Initialize:** Policy and value networks $\pi_\theta, V_\phi$, hyperparameter $\epsilon$, value loss and policy entropy coefficients $c_1, c_2$
2: **Output:** Optimal policy network $\pi_{\theta*}$
3: **while** not converged **do**
4:     # Data collection
5:     Collect data $\mathcal{D} = \{(s_t, a_t, r_t)\}_{t=1}^N$ using the current policy network $\pi_\theta$
6:     # The networks before updating
7:     $\pi_{\theta_{\text{old}}} \leftarrow \pi_\theta, \ V_{\phi_{\text{old}}} \leftarrow V_\phi$
8:     # Estimate the advantage $\hat{A}(s_t, a_t)$ based on $V_{\phi_{\text{old}}}$
9:     Use GAE (Schulman et al., 2015b) technique to estimate the advantage $\hat{A}(s_t, a_t)$
10:     # Estimate the return $\hat{R}_t$
11:     $\hat{R}_t \leftarrow V_{\phi_{\text{old}}}(s_t) + \hat{A}(s_t, a_t)$
12:     **for** each training epoch **do**
13:         # Compute policy loss $\mathcal{L}_p$ (This is the only difference between SPO and PPO)
14:         $\mathcal{L}_p \leftarrow -\frac{1}{N} \sum_{t=1}^N \left\{ \frac{\pi_\theta(a_t|s_t)}{\pi_{\theta_{\text{old}}}(a_t|s_t)} \cdot \hat{A}(s_t, a_t) - \frac{|\hat{A}(s_t, a_t)|}{2\epsilon} \cdot \left[ \frac{\pi_\theta(a_t|s_t)}{\pi_{\theta_{\text{old}}}(a_t|s_t)} - 1 \right]^2 \right\}$
15:         # Compute policy entropy $\mathcal{L}_e$ and value loss $\mathcal{L}_v$
16:         $\mathcal{L}_e \leftarrow \frac{1}{N} \sum_{t=1}^N \mathcal{H}(\pi_\theta(\cdot|s_t)), \ \mathcal{L}_v \leftarrow \frac{1}{2N} \sum_{t=1}^N [V_\phi(s_t) - \hat{R}_t]^2$
17:         # Compute total loss $\mathcal{L}$
18:         $\mathcal{L} \leftarrow \mathcal{L}_p + c_1 \mathcal{L}_v - c_2 \mathcal{L}_e$
19:         # Update parameters $\theta$ and $\phi$ through backpropagation, $\lambda_\theta$ and $\lambda_\phi$ is the step sizes
20:         $\theta \leftarrow \theta - \lambda_\theta \nabla_\theta \mathcal{L}, \ \phi \leftarrow \phi - \lambda_\phi \nabla_\phi \mathcal{L}$
21:     **end for**
22: **end while**

---

Based on the Proposition 4.2 and Theorem 4.3, we have the following conclusion:

> **Conclusion**
>
> Optimizing the lower bound with TV divergence constrains offers a more effective solution space than using KL divergence constrains, leading to better policy improvement.

As a result, to optimize the lower bound (10), we aim to solve the following constrained optimization problem:

$$\max_\theta \ \mathbb{E}_{(s_t,a_t)\sim\pi_{\theta_{\text{old}}}} \left[ r_t(\theta) \cdot \hat{A}_t \right],$$
$$\text{s.t.} \ \mathbb{E}_{(s_t,a_t)\sim\pi_{\theta_{\text{old}}}} \left[ |r_t(\theta) - 1| \right] \le \epsilon, \quad (15)$$

where $r_t(\theta) = \pi_\theta(a_t|s_t)/\pi_{\theta_{\text{old}}}(a_t|s_t)$ and $\hat{A}_t = \hat{A}(s_t, a_t)$.

PPO attempts to satisfy the constraints of (15) through ratio clipping, but this does not prevent excessive ratio deviations (demonstrated in Figure 2). The underlying reason is that ratio clipping causes certain data points to stop contributing to the gradients. Over multiple iterations, this can lead to uncontrollable updates, as the absence of corrective gradients prevents the policy from recovering. To overcome this issue

with ratio clipping, we propose the following objective:

$$J(\theta) = \mathbb{E}_{(s_t,a_t)\sim\pi_{\theta_{\text{old}}}} \left\{ r_t(\theta) \cdot \hat{A}_t - \frac{|\hat{A}_t|}{2\epsilon} \cdot [r_t(\theta) - 1]^2 \right\}. \quad (16)$$

The details of the objective will be discussed in the following section, and the pseudo-code is shown in Algorithm 1.

## 5. Theoretical Results

In this section, we provide some theoretical insights of the differences between PPO and SPO, demonstrating that SPO can be more effective in constraining probability ratios.

### 5.1. Objective Class

Simplify the notation by using $r$ and $A$ to represent the probability ratio and the advantage value. Based on the previous analysis, our goal is to find an objective function $f(r, A, \epsilon)$ such that while optimizing the surrogate objective $rA$, the probability ratio is constrained by $|r - 1| \le \epsilon$.

According to (15), for any given $A \ne 0$ and $\epsilon > 0$, we can write down the following desired optimization problem:

$$\max_r \ rA, \ \text{s.t.} \ |r - 1| \le \epsilon. \quad (17)$$

The objective is linear, so the optimal solution is $r^* = 1 + \text{sign}(A) \cdot \epsilon$, where $\text{sign}(\cdot)$ is the sign function. Motivated by this, we present the following definition:

**Definition 5.1** ($\epsilon$-aligned). For any given $A \neq 0$ and $\epsilon > 0$, we say that the function $f(r, A, \epsilon)$ is $\epsilon$-aligned, if it is differentiable and convex with respect to $r$, and attains its maximum value at $r = 1 + \text{sign}(A) \cdot \epsilon$.

The objective of PPO in (6) and SPO in (16) can be expressed as

$$f_{\text{ppo}} = \min\left[rA, \text{clip}(r, 1 - \epsilon, 1 + \epsilon)A\right],$$
$$f_{\text{spo}} = rA - \frac{|A|}{2\epsilon} \cdot (r - 1)^2. \qquad (18)$$

It can be obtained that $f_{\text{ppo}}$ is not $\epsilon$-aligned, as $f_{\text{ppo}}$ zeros the gradients under some special cases according to (8). For $f_{\text{spo}}$, we have the following theorem:

**Theorem 5.2.** $f_{\text{spo}}$ is $\epsilon$-aligned.

*Proof.* Obviously, $f_{\text{spo}}$ is differentiable and convex with respect to $r$ since $f_{\text{spo}}$ is a quadratic polynomial of $r$. For any given $A \neq 0$ and $\epsilon > 0$, let $\partial f_{\text{spo}}(r, A, \epsilon)/\partial r = 0$, then

$$\frac{\partial f_{\text{spo}}(r, A, \epsilon)}{\partial r} = A - \frac{|A|}{\epsilon} \cdot (r - 1) = 0, \qquad (19)$$

thus $r = 1 + \text{sign}(A) \cdot \epsilon$ is the optimal solution for $f_{\text{spo}}$. $\square$

Note that $f_{\text{spo}}$ is not the only objective function that satisfies the definition. It can be proved that there is a simple objective function $f_{\text{simple}} = -(r - 1 - \text{sign}(A) \cdot \epsilon)^2$, which is also $\epsilon$-aligned. We will discuss the differences between these two in Section 6.3.

### 5.2. Analysis of New Objective

We show that the optimization process of SPO can more effectively bound the probability ratio, as can be seen from Figure 3. The largest circular area in the figure represents the boundary on the probability ratio. The green circles represent data points with non-zero gradients during the training process, while the gray circles represent data points with zero gradients.

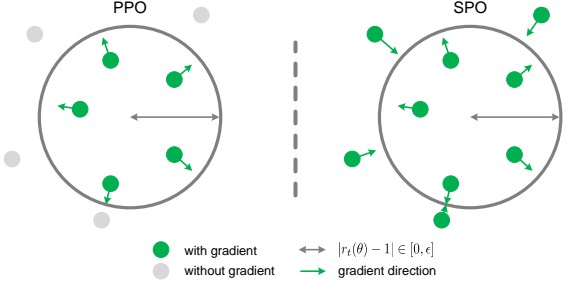

*Figure 3.* In PPO, certain data points exhibit zero gradients, while in SPO, all data points generate non-zero gradients that guide towards the constraint boundary.

During the training process of PPO, certain data points that exceed the probability ratio bound cease to provide gradients. In contrast, all data points in SPO contribute gradients that guide the optimization towards the constraint boundary. As training progresses, PPO will accumulate more gray circles that no longer provide gradients and may be influenced by the harmful gradients from green circles. This phenomenon could potentially push the gray circles further away from the constraint boundary. In contrast, the gradient directions of all data points in SPO point towards the constraint boundary. This indicates that SPO imposes stronger constraints on the probability ratio.

## 6. Experiments

We report results on the Atari 2600 (Bellemare et al., 2013; Machado et al., 2018) and MuJoCo (Todorov et al., 2012) benchmarks. In all our experiments, we utilize the RL library Gymnasium (Towers et al., 2024), which serves as a central abstraction to ensure broad interoperability between benchmark environments and training algorithms.

### 6.1. Comparing Algorithms

Our implementation of SPO is compared against PPO-Clip (Schulman et al., 2017), PPO-Penalty (Schulman et al., 2017), SPU (Vuong et al., 2018), PPO-RB (Wang et al., 2020), TR-PPO (Wang et al., 2020), TR-PPO-RB (Wang et al., 2020), and RPO (Gan et al., 2024) in MuJoCo benchmark. We compute the algorithm's performance across ten separate runs with different random seeds. In addition, we emphasize that in all comparative experiments involving the same settings for SPO and PPO, the only modification in SPO is replacing the PPO's objective with (16), no further code-level tuning is applied to SPO, highlighting its simplicity and efficiency.

Due to the absence of human score baselines in MuJoCo (Todorov et al., 2012), we normalize the algorithms' performance across all environments using the training data of PPO-Clip, specifically,

$$\text{normalized(score)} = \frac{\text{score} - \min}{\max - \min}, \qquad (20)$$

where $\max$ and $\min$ represent the maximum and minimum validation returns of PPO-Clip during training, respectively.

As suggested in Agarwal et al. (2021), we employ stratified bootstrap confidence intervals to assess the confidence intervals of the algorithm and evaluate the composite metrics of SPO against other baselines, as illustrated in Figure 4. It can be observed that SPO achieved the best performance across nearly all statistical metrics, which fully demonstrates the strong potential of SPO. For the Atari 2600 benchmark (Bellemare et al., 2013), the main results are presented in Appendix A and C.

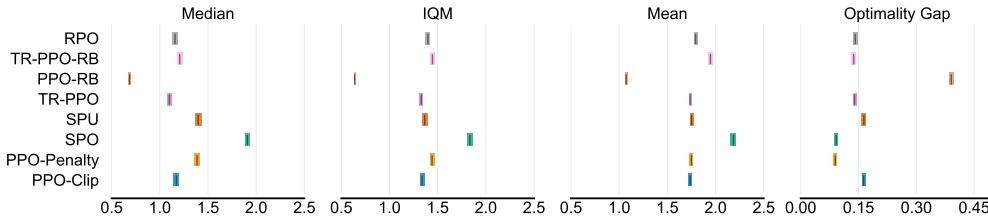

*Figure 4.* Aggregate metrics on MuJoCo-v4 with 95% CIs based on 6 environments. We collected the returns of each algorithm over the last 1% training steps across ten random seeds. In this context, higher median, IQM and mean scores and lower optimality gap are better.

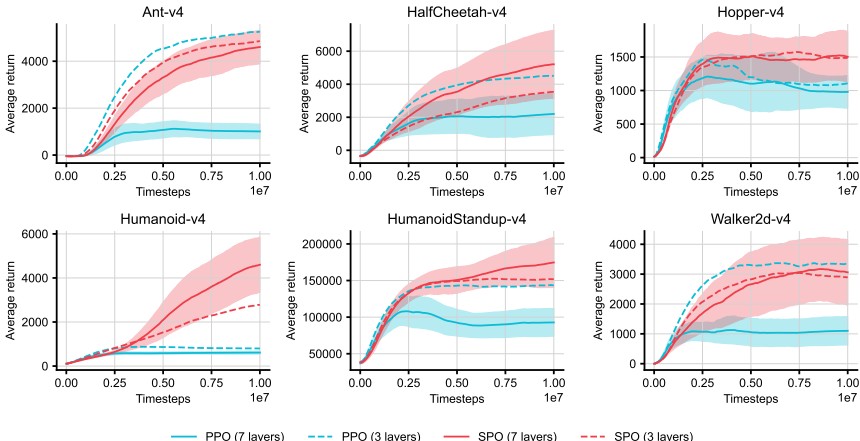

*Figure 5.* Training performance of PPO and SPO with different policy network layers in MuJoCo benchmark. The mean and standard deviation are shown across 5 random seeds.

*Table 1.* Average return of PPO and SPO in the last 10% training steps across 5 separate runs with different random seeds, with their maximum ratio deviation during the entire training process.

| Environment | Index | 3 layers | | 7 layers | |
|---|---|---|---|---|---|
| | | PPO | SPO | PPO | SPO |
| Ant-v4 | Average return (↑) | **5323.2** | 4911.3 | 1002.8 | **4672.5** |
| | Ratio deviation (↓) | 0.229 | **0.101** | 548.060 | **0.190** |
| HalfCheetah-v4 | Average return (↑) | **4550.2** | 3602.4 | 2242.3 | **5307.3** |
| | Ratio deviation (↓) | 0.225 | **0.086** | 1675.340 | **0.188** |
| Hopper-v4 | Average return (↑) | 1119.4 | **1480.3** | 975.9 | **1507.6** |
| | Ratio deviation (↓) | 0.164 | **0.067** | 113.178 | **0.194** |
| Humanoid-v4 | Average return (↑) | 795.1 | **2870.0** | 614.1 | **4769.9** |
| | Ratio deviation (↓) | 3689.957 | **0.179** | 2411.845 | **0.191** |
| HumanoidStandup-v4 | Average return (↑) | 143908.8 | **152378.7** | 92849.7 | **176928.9** |
| | Ratio deviation (↓) | 2547.499 | **0.182** | 4018.718 | **0.187** |
| Walker2d-v4 | Average return (↑) | **3352.3** | 2870.2 | 1110.9 | **3008.1** |
| | Ratio deviation (↓) | 0.170 | **0.070** | 998.101 | **0.157** |

### 6.2. Scaling Policy Network

To investigate how scaling policy network size impacts the sample efficiency of both PPO and SPO in MuJoCo, the number of policy network layers was increased without altering the hyperparameters or other settings. The standard deviation of the algorithm's performance was computed and visualized across five separate runs with different random seeds. The results, shown in Figure 5, 9 and Table 1, where the *ratio deviation* indicates the largest value of average

ratio deviation in a batch during the entire training process, i.e., $\frac{1}{|\mathcal{D}|} \sum_{(s_t,a_t)\sim\mathcal{D}} |r_t(\theta) - 1|$.

It can be observed that as the network deepens, the performance of PPO collapses in most environments, with uncontrollable probability ratio deviations. In contrast, the performance of SPO outperforms that of shallow networks in almost all environments and constrains the probability ratio deviation effectively. Furthermore, the statistical metrics of SPO generally outperform PPO's and demonstrate

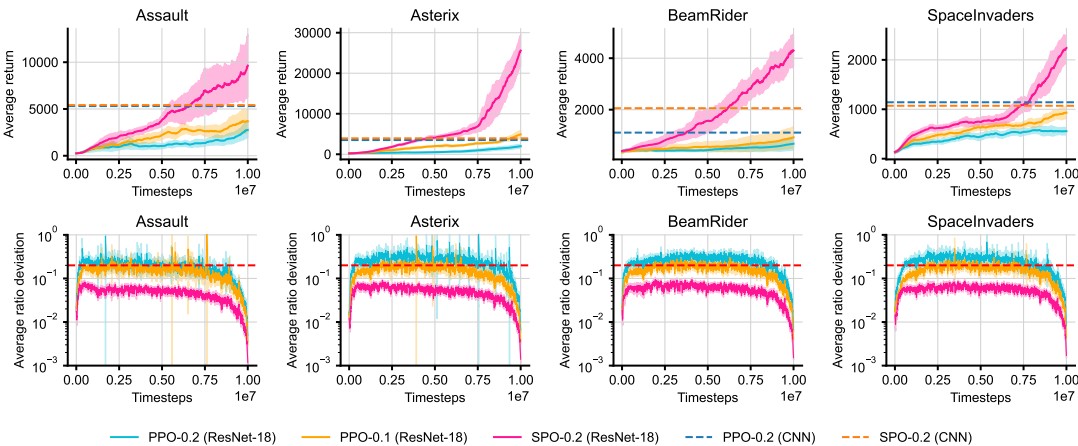

Figure 6. Training performance of SPO using ResNet-18 as the encoder compared to the PPO and SPO using default CNN (shown with the reference line). The mean and standard deviation are shown across 3 random seeds, and the red dashed line represents $\epsilon = 0.2$.

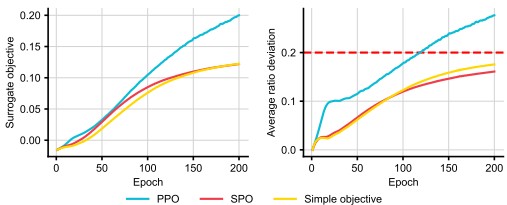

Figure 7. The optimization behavior of $f_{\text{ppo}}$, $f_{\text{spo}}$, and $f_{\text{simple}}$.

relative robustness to variations in network depth and mini-batch size.

We also trained the ResNet-18[1] (He et al., 2016) as the encoder on the Atari 2600 benchmark, the results are shown in Figure 6. As the network's capacity increases, the performance of SPO is significantly improved. Moreover, SPO can still maintain a good probability ratio constraint, thereby benefiting from the theoretical lower bound (10). In contrast, it is challenging to train large neural networks with PPO because the probability ratio cannot be controlled during training, even employing a smaller $\epsilon = 0.1$.

### 6.3. Constraining Ratio Deviation

To further investigate the optimization behavior of different objective functions that satisfy the $\epsilon$-aligned definition, we visualize the optimization process of $f_{\text{ppo}}$, $f_{\text{spo}}$, and $f_{\text{simple}}$ presented in Section 5.1, on the same batch of advantage values initialized from a standard Gaussian distribution, as shown in Figure 7.

We can observe that while PPO achieves the best perfor-

mance in optimizing the surrogate objective, it also leads to uncontrollable ratio deviations. In contrast, the two objectives that satisfy the $\epsilon$-aligned definition effectively constrain the ratio deviations during the optimization process.

Furthermore, we also observe that $f_{\text{spo}}$ achieves better optimization of the surrogate objective compared to $f_{\text{simple}}$, while $f_{\text{simple}}$ converges more quickly to the probability ratio boundary. This aligns with our expectations, as the optimization objective of $f_{\text{simple}}$ only depends on the sign of the advantage values. As a result, $f_{\text{simple}}$ pushes each data point equally toward the constraint boundary, which results in the magnitude of the advantage values being less effectively utilized compared to $f_{\text{spo}}$, which makes it difficult to efficiently optimize the surrogate objective.

## 7. Conclusion

In this paper, we introduce *Simple Policy Optimization* (SPO), a novel unconstrained first-order algorithm that effectively combines the strengths of Trust Region Policy Optimization (TRPO) and Proximal Policy Optimization (PPO). SPO maintains optimization within the trust region, benefiting from TRPO's theoretical guarantees while preserving the efficiency of PPO. Our experimental results demonstrate that SPO achieves competitive performance across various benchmarks with a simple implementation. Moreover, SPO simplifies the training of deep policy networks, addressing a key challenge faced by existing algorithms. These findings indicate that SPO is a promising approach for advancing model-free reinforcement learning. In future work, SPO holds potential for impactful applications in areas such as language models, robotic control, and financial modeling. With further research and refinement, we believe SPO will drive innovation and breakthroughs across these fields.

---

[1]Since Bhatt et al. (2019) demonstrated that batch normalization is harmful to RL training, we removed batch normalization.

## Acknowledgements

We would like to thank the anonymous ICLR and ICML reviewers for their insightful and constructive comments.

## Impact Statement

This paper presents work whose goal is to advance the field of Machine Learning. There are many potential societal consequences of our work, none which we feel must be specifically highlighted here.

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

## A. Atari 2600

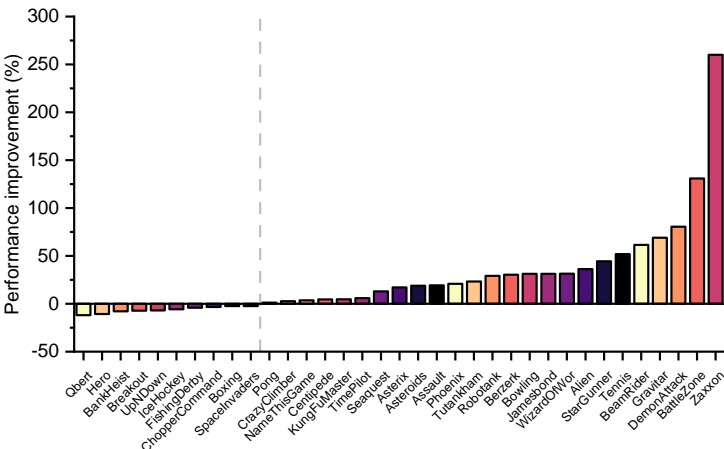

*Figure 8.* Final performance of SPO compared to PPO across 35 games in Atari 2600 environment, using default CNN as the encoder.

## B. Hyperparameters

*Table 2.* Detailed hyperparameters used in SPO.

| Hyperparameters | Atari 2600 (Bellemare et al., 2013) | MuJoCo (Todorov et al., 2012) |
|---|---|---|
| Number of workers | 8 | 8 |
| Horizon | 128 | 256 |
| Learning rate | 0.00025 | 0.0003 |
| Learning rate decay | Linear | Linear |
| Optimizer | Adam | Adam |
| Total steps | 10M | 10M |
| Batch size | 1024 | 2048 |
| Update epochs | 4 | 10 |
| Mini-batches | 4 | 4 |
| Mini-batch size | 256 | 512 |
| GAE parameter $\lambda$ | 0.95 | 0.95 |
| Discount factor $\gamma$ | 0.99 | 0.99 |
| Value loss coefficient $c_1$ | 0.5 | 0.5 |
| Entropy loss coefficient $c_2$ | 0.01 | 0.0 |
| Probability ratio hyperparameter $\epsilon$ | 0.2 | 0.2 |

## C. More Results

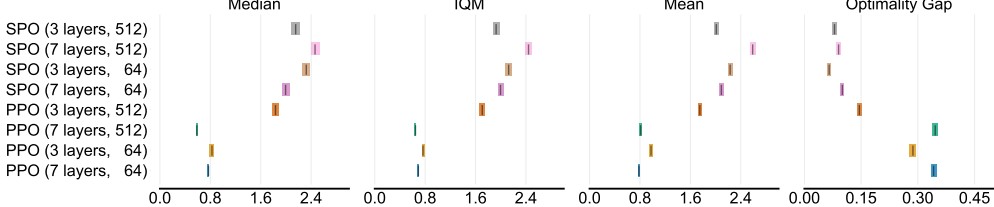

*Figure 9.* Aggregate metrics on MuJoCo-v4 with 95% CIs based on 6 environments, comparing PPO and SPO with different policy network layers and mini-batch sizes using PPO-normalized score.

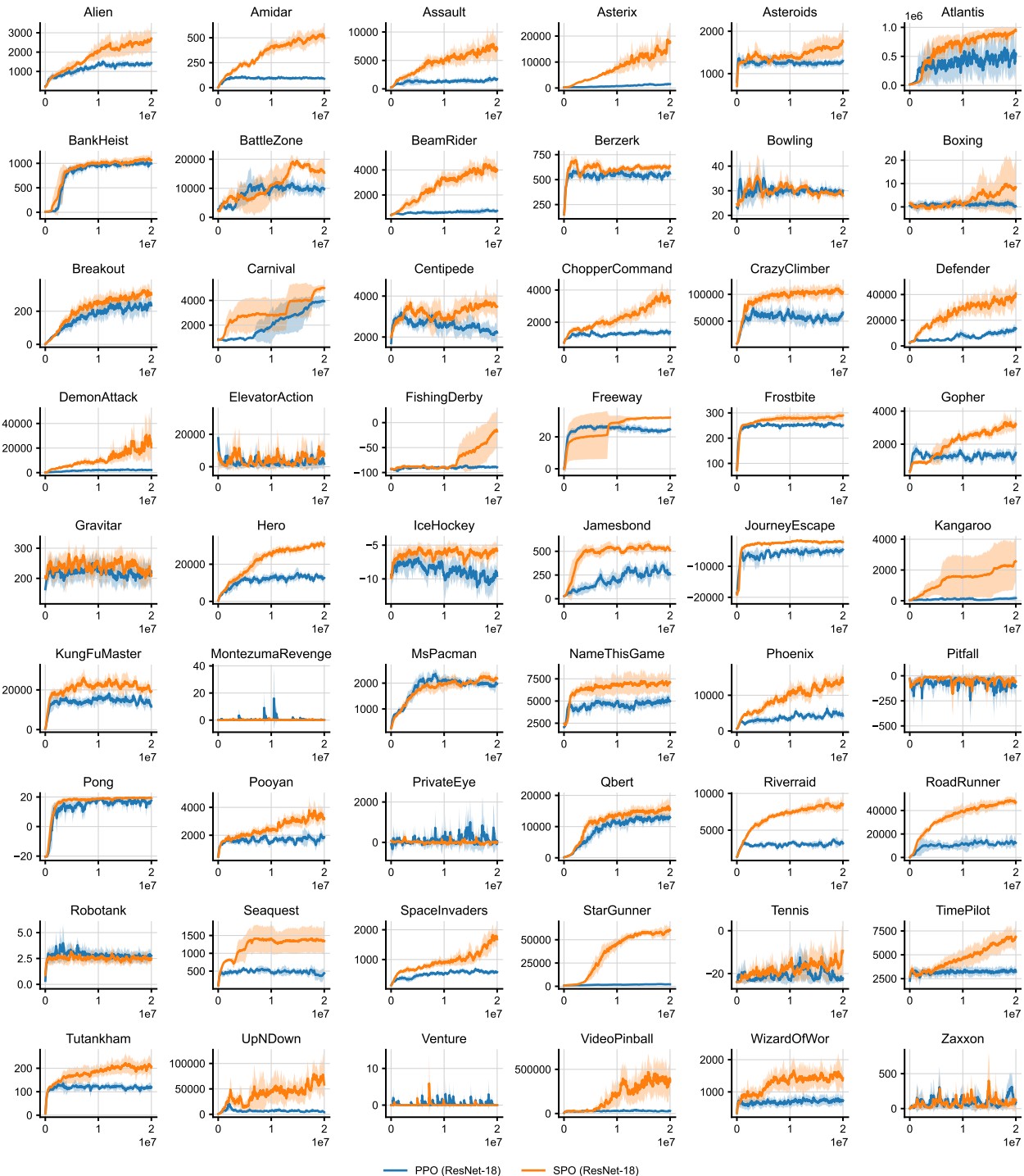

*Figure 10.* Training performance on Atari 2600 using ResNet-18 as the encoder, with a fixed learning rate of 0.0001. The mean and standard deviation are shown across 3 random seeds.

