# OpenReview forum: "Simple Policy Optimization"
_ICML.cc/2025/Conference — ICML 2025 poster_

### Official Review · Reviewer_uY8N · 2025-03-12

**Overall Recommendation:** 3

**Summary:**

This paper theoretically identifies a flaw of the clipping technique in PPO's objective and proposes a solution to it. Empirical results show that the proposed solution achieves comparable or better performance on MuJoCo and Atari, and the performance improves as the policy network scales.

**Claims And Evidence:**

The abstract claims that TRPO has a strong theoretical foundation while PPO has better practical efficiency, and that the proposed method achieves the best of both worlds. This sounds a bit odd because from the empirical results the proposed method actually performs better than PPO (this is even stated at the end of the abstract).

**Essential References Not Discussed:**

N/A

**Experimental Designs Or Analyses:**

Figure 10: The results on Atari are averaged over only 3 random seeds, which I think is not sufficient.

**Methods And Evaluation Criteria:**

Yes.

**Other Comments Or Suggestions:**

The results in Figure 8 look stronger than the MuJoCo results and would be better if put in the main paper instead of the appendix.

**Other Strengths And Weaknesses:**

- The theoretical explanation of why PPO's clipping technique may not be optimal is convincing, and the corresponding illustration in Figure 2 is intuitive.
- The implementation of the proposed method is simple.
- The empirical performance of the proposed method is strong, especially on Atari tasks. The fact that the performance improves as the network scales is particularly promising.

**Questions For Authors:**

- Could you specify which implementation of PPO you are using (which implementation SPO is based on)? I am a little concerned since as also pointed out by this paper, existing work has shown that the performance of PPO is highly dependent on code-level optimization. It would be better if the url of the implementation is included in the paper.
- Have you tried scaling the width of the network besides the depth? I'm curious because a recent paper has shown that scaling the width can yield significant performance improvements on its own \[1\].
- Do you have any idea why PPO tends to perform worse and lose control of the probability ratio as the network scales?
- Could you show the scaling performance in Atari? It would be more convincing if there are multiple benchmarks where the results support the claim.
- Figure 2 (right): Could you specify the task on which the results are produced?



\[1\] Obando-Ceron et al., "In value-based deep reinforcement learning, a pruned network is a good network".

**Relation To Broader Scientific Literature:**

N/A

**Theoretical Claims:**

No, I did not.

---

> ### Author Rebuttal · Authors · 2025-04-01
>
> Dear Reviewer uY8N,
>
> Thank you for your positive feedback. Below, we will address your concerns.
>
> >Figure 10: The results on Atari are averaged over only 3 random seeds, which I think is not sufficient.
>
> Thank you for your suggestion. Given the high computational costs associated with Atari environments, the use of 3 random seeds represents a well-established practice in the field [2]. Furthermore, due to time constraints during the rebuttal period, we regret that we are unable to conduct additional experiments with more random seeds.
>
> >The results in Figure 8 look stronger than the MuJoCo results and would be better if put in the main paper instead of the appendix.
>
> Thank you for your suggestion! Due to space constraints, we have included the key content in the main text (Figure 4 compares a broader range of baselines). If the paper is accepted, we will also incorporate Figure 8 into the main body.
>
> >Could you specify which implementation of PPO you are using (which implementation SPO is based on)?...
>
> Our implementation of PPO is based on the standard CleanRL library: https://github.com/vwxyzjn/cleanrl, with the only modification being the computation of the policy loss in SPO. The complete code for this study has been submitted as part of the **supplementary materials**.
>
> >Have you tried scaling the width of the network besides the depth? I'm curious because a recent paper has shown that scaling the width can yield significant performance improvements on its own [1].
>
> Thank you for the additional paper. Our three-layer network follows the default settings: [$\mathrm{dim}(\mathcal{S})$, 64, 64, $\mathrm{dim}(\mathcal{A})$], while the seven-layer network is both deeper and wider in architecture: [$\mathrm{dim}(\mathcal{S})$, 256, 256, 128, 128, 64, 64, $\mathrm{dim}(\mathcal{A})$].
>
> >Do you have any idea why PPO tends to perform worse and lose control of the probability ratio as the network scales?
>
> Certainly! We'd be happy to provide a simplified explanation. Generally speaking, deeper networks tend to have larger Lipschitz constants, meaning that even minor changes in the network's input can lead to significant variations in its output. Consequently, when using deeper architectures, PPO's limitations become more pronounced, as slight parameter adjustments may cause the probability ratio to exceed its boundary.
>
> >Could you show the scaling performance in Atari? It would be more convincing if there are multiple benchmarks where the results support the claim.
>
> Thank you for your suggestion. We primarily demonstrate the scaling performance in Atari environments through Figures 1 and 6. Specifically, Figure 1 shows that SPO can effectively train ResNets with over 100 layers, while Figure 6 highlights SPO's significant performance improvements using ResNet-18 across four Atari environments.
>
> Additionally, due to the limited rebuttal period, we may require additional time to obtain the Atari results, potentially during the discussion phase.
>
> >Figure 2 (right): Could you specify the task on which the results are produced?
>
> Sure! We generated Figure 2 using the Hopper-v4 environment. In fact, similar results can be reproduced in any environment (e.g., Humanoid-v4).
>
> Best,
>
> Authors
>
> ---
> *Reference:*
>
> [1] J Obando-Ceron et al. In value-based deep reinforcement learning, a pruned network is a good network.
>
> [2] Y Gan et al. Reflective policy optimization.

---

### Official Review · Reviewer_eeAe · 2025-03-13

**Overall Recommendation:** 3

**Summary:**

This paper introduces Simple Policy Optimization (SPO), a first-order algorithm that modifies PPO's policy loss to achieve stronger theoretical properties, particularly in bounding the probability ratios between successive policies. The authors argue that by optimizing a lower bound under TV divergence constraints, SPO provides a more effective solution space than approaches using KL divergence (e.g., TRPO). Empirical results show that SPO performs comparably to PPO across some benchmarks.

**Claims And Evidence:**

Partially. The claim regarding improved theoretical properties is plausible, but empirical support for SPO’s superiority over PPO and TRPO is limited.

**Essential References Not Discussed:**

No.

**Experimental Designs Or Analyses:**

Experimental designs are reasonable but lack robustness due to a small number of seeds and limited hyperparameter tuning discussions.

**Methods And Evaluation Criteria:**

Yes, the methods are appropriate for policy optimization tasks, though additional experiments would strengthen the evaluation.

**Other Comments Or Suggestions:**

- In the experiments, provide more details on architectural choices (e.g., why 3-layer and 7-layer networks were compared). Discuss whether the variations in performance are due to SPO or other factors.
- Include a comparison with TRPO, as the paper suggests that SPO combines the advantages of both TRPO and PPO. A head-to-head comparison would strengthen this claim.
- Provide clearer explanations in the captions of Figures 5, 6, and 7. For example, explain what the numeric values in the legends (e.g., 0.1, 0.2) represent and clarify the meaning of horizontal lines.
- Expand the experimental evaluation to include more random seeds (at least 10) to enhance the reliability of the results.

**Other Strengths And Weaknesses:**

**Strengths**

- Proposes a modification to the PPO objective that aims to provide stronger theoretical guarantees.
- Provides theoretical analysis comparing different divergence constraints and their impact on policy optimization.
- Experimental validation includes several benchmark environments, which help illustrate the method’s potential in diverse tasks.

**Weaknesses**

- The paper lacks a clear demonstration of significant performance improvements over PPO, with most experimental results showing only marginal gains.
- Claims regarding the theoretical advantages of SPO over PPO and TRPO are not fully substantiated with rigorous empirical validation.
- Key implementation details and hyperparameters (e.g., the reasoning behind comparing 3-layer vs. 7-layer architectures) are insufficiently explained. This makes it difficult to assess whether the results stem from algorithmic improvements or implementation choices.
- Several experimental setups (e.g., Figure 6) lack clarity in terms of legend explanations and parameter settings. Additionally, the small number of random seeds (only three) undermines the statistical robustness of the results.

**Questions For Authors:**

Q1.  How do the proposed SPO results compare directly with TRPO across the same benchmarks? Can you provide experimental comparisons to clarify this?

Q2.  Is there a tunable parameter in SPO that allows interpolation between PPO-like and TRPO-like behavior? If so, how does varying this parameter affect performance?

Q3.  What is the meaning of the values shown in the legends of Figures 5 and 6 (e.g., 0.1, 0.2)? If these values represent $\epsilon$, how is  $\epsilon$ specifically related to the ResNet architecture and not to other settings? What do the horizontal lines in these figures represent?

Q4.  Can you clarify the meaning of Theorem 5.1 (that SPO is $\epsilon$-aligned)? How does this translate into practical advantages in policy optimization?

**Relation To Broader Scientific Literature:**

Builds upon PPO and TRPO, addressing known issues with PPO's trust region enforcement.

**Theoretical Claims:**

While the high-level ideas of the theorem seem sound, the integration into the overall argument could be clearer.

---

> ### Author Rebuttal · Authors · 2025-04-01
>
> Dear Reviewer eeAe,
>
> Thank you for your constructive feedback. Below, we will address your concerns.
>
> >Partially. The claim regarding improved theoretical properties is plausible, but empirical support for SPO’s superiority over PPO and TRPO is limited.
>
> >The paper lacks a clear demonstration of significant performance improvements over PPO, with most experimental results showing only marginal gains.
>
> >Include a comparison with TRPO, as the paper suggests that SPO combines the advantages of both TRPO and PPO. A head-to-head comparison would strengthen this claim.
>
> Thank you for your suggestion. However, as demonstrated in Figures 1, 4, and 5 of our paper, our experiments show that SPO indeed exhibits significant advantages over PPO. Notably, Figure 4 includes comparisons with **a wide range of baselines**. Unfortunately, due to TRPO's computationally expensive second-order optimization, it is typically not used as a baseline in existing works [1, 2, 3].
>
> >Claims regarding the theoretical advantages of SPO over PPO and TRPO are not fully substantiated with rigorous empirical validation.
>
> Thank you for your feedback. As shown in Table 1 of our paper, when PPO trains deeper networks, the probability ratio becomes uncontrolled and leads to performance collapse in PPO. Please note that in this experiment, the only variable modified was the network depth - no additional code-level tuning was performed for SPO. We believe this strongly demonstrates SPO's theoretical advantage, as its probability ratio remains effectively constrained without performance degradation even with increasing network depth.
>
> >Key implementation details and hyperparameters (e.g., the reasoning behind comparing 3-layer vs. 7-layer architectures) are insufficiently explained...
>
> >In the experiments, provide more details on architectural choices (e.g., why 3-layer and 7-layer networks were compared)...
>
> Thank you for your suggestion. The reviewer's concern primarily relates to the fairness of the experimental setup and the potential possibility that SPO might involve additional hyperparameters or code-level tuning. Regarding this point, please refer to our already uploaded SPO and PPO implementations, where you will find that the only difference between SPO and PPO lies in the policy loss computation (the sole modification in the trainer.py file, with all other settings remaining identical).
>
> As for the network depth selection, we employed both the default three-layer mlp [$\mathrm{dim}(\mathcal{S})$, 64, 64, $\mathrm{dim}(\mathcal{A})$] and a randomly selected seven-layer mlp [$\mathrm{dim}(\mathcal{S})$, 256, 256, 128, 128, 64, 64, $\mathrm{dim}(\mathcal{A})$] to ensure comprehensive evaluation.
>
> >Several experimental setups (e.g., Figure 6) lack clarity in terms of legend explanations and parameter settings. Additionally, the small number of random seeds (only three) undermines the statistical robustness of the results.
>
> Thank you for your careful review. In Figure 6, the red dashed line represents the value of 0.2, while all other elements are clearly indicated in the legend.
>
> Regarding the random seeds, due to the computational costs in Atari environments, using 3 seeds is a common choice [4]. Furthermore, for the MuJoCo environments, the results presented in Figure 4 were indeed obtained using 10 random seeds to ensure statistical reliability.
>
> >Q2. Is there a tunable parameter in SPO that allows interpolation between PPO-like and TRPO-like behavior?...
>
> Please note that TRPO and PPO exhibit fundamentally different optimization behaviors. SPO combines TRPO's theoretical guarantee (monotonic improvement) with PPO's computational efficiency (eliminating the need for second-order optimization).
>
> >Q3. What is the meaning of the values shown in the legends of Figures 5 and 6 (e.g., 0.1, 0.2)?...
>
> The red dashed lines in these figures represent the hyperparameter epsilon (typically set to 0.2), and we will add the corresponding legend in subsequent revisions. The remaining horizontal lines, which are already labeled in the legend, indicate the performance of the original CNN at convergence (serving as baseline references).
>
> >Q4. Can you clarify the meaning of Theorem 5.1 (that SPO is $\epsilon$-aligned)? How does this translate into practical advantages in policy optimization?
>
> Please refer to the performance improvement lower bound (10). Theorem 5.1 demonstrates that SPO can indirectly optimize this lower bound (10) by constraining the probability ratio deviation $\|\frac{\tilde{\pi}(a|s)}{\pi(a|s)}-1\|\leq\epsilon$.
>
>
>
>
> Best,
>
> Authors
>
> ---
> *Reference:*
>
> [1] K Cobbe et al. Leveraging procedural generation to benchmark reinforcement learning.
>
> [2] Y Wang et al. Truly proximal policy optimization.
>
> [3] Y Wang et al. Trust region-guided proximal policy optimization.
>
> [4] Y Gan et al. Reflective policy optimization.

---

> > ### Comment · Reviewer_eeAe · 2025-04-03
> >
> > Thanks to the authors for the response.
> >
> > Reiterating my concerns regarding architecture choice:
> > What is the rationale behind comparing 3-layer versus 7-layer architectures? I’m still unclear on why a particular choice of architecture might favor SPO over others—either from a theoretical or empirical standpoint. For instance, in Figure 5 (Ant-v4), PPO with a 3-layer architecture outperforms all other settings, including SPO with both 3 and 7 layers. However, SPO with 7 layers performs better than PPO with 7 layers. Similar trends appear in other environments depicted in Figure 5. While these are certainly interesting empirical findings, is there any theoretical justification or further insight into why SPO might perform better under certain architectural configurations?

---

> > > ### Author Response · Authors · 2025-04-04
> > >
> > > Dear Reviewer eeAe,
> > >
> > > Thank you for your additional comments. Below, we will address your concerns.
> > >
> > > We did not deliberately choose the network architecture or depth; the primary motivation of the experiment was to reveal that PPO **fails to constrain the probability ratio**. To explain why network depth (or, more generally, network complexity) can make this phenomenon more pronounced, we provide the following insights:
> > >
> > > According to the empirical results in [1], the default policy network used in the MuJoCo is [$\mathrm{dim}(\mathcal{S})$, 64, 64, $\mathrm{dim}(\mathcal{A})$], which is a network structure with fairly limited capacity. As the network becomes deeper (or wider), small changes in parameters can lead to large variations in output. When training neural networks with a larger number of parameters, PPO’s clipping mechanism causes some data to have zero gradients. Due to the large number of parameters, the proportion of data that actually contributes to gradients during PPO's training **decreases faster compared to shallower networks**. This leads to larger bias in the data that provides gradients, ultimately pushing the policy entirely out of the trust region and resulting in performance collapse.
> > >
> > > SPO addresses this issue because each data point in SPO provides a gradient directed toward the constraint boundary. As a result, data points that attempt to escape the trust region are pulled back by the gradient, thereby enforcing the trust region constraint more effectively and leading to stable performance improvements.
> > >
> > > To illustrate this, we further conducted the following experiment, where the policy network was set to
> > >
> > > [$\mathrm{dim}(\mathcal{S})$, 256, 256, 256, $\mathrm{dim}(\mathcal{A})$] and [$\mathrm{dim}(\mathcal{S})$, 512, 512, 512, $\mathrm{dim}(\mathcal{A})$]
> > >
> > > to rule out the possibility that SPO's performance benefits from a specific network architecture. The results are as follows:
> > >
> > > **Policy network: [$\mathrm{dim}(\mathcal{S})$, 256, 256, 256, $\mathrm{dim}(\mathcal{A})$]**
> > > | Algorithm   | Ant-v4 | Humanoid-v4  | HumanoidStandup-v4 |
> > > |--------|-----|------|------|
> > > | PPO   | $-58.73\pm57.06$| $513.8\pm49.02$| $72210.81\pm10491.1$ |
> > > | SPO   | $4048.64\pm1045.12$| $2504.07\pm981.37$ |$149694.08\pm20166.95$ |
> > >
> > > **Policy network: [$\mathrm{dim}(\mathcal{S})$, 512, 512, 512, $\mathrm{dim}(\mathcal{A})$]**
> > > | Algorithm   | Ant-v4 | Humanoid-v4  | HumanoidStandup-v4 |
> > > |--------|-----|------|------|
> > > | PPO   | $-36.58\pm31.42$| $580.09\pm39.65$| $77964.68\pm14870.27$ |
> > > | SPO   | $2278.72\pm751.46$| $1971.7\pm919.23$ |$155631.11\pm23507.7$ |
> > >
> > > We can see that as the network parameter increases, PPO can not learn a good policy, whereas SPO is still able to perform well. We also demonstrate the **average and maximum ratio deviation** of PPO and SPO during the training process under these two network structures:
> > >
> > > **Policy network: [$\mathrm{dim}(\mathcal{S})$, 256, 256, 256, $\mathrm{dim}(\mathcal{A})$]**
> > > | Algorithm   | Ant-v4 | Humanoid-v4  | HumanoidStandup-v4 |
> > > |--------|-----|------|------|
> > > | PPO   | $6.31(4264.61)$| $24.15(22444.92)$| $28.77(34402.84)$ |
> > > | SPO   | $0.16(0.29)$| $0.16(0.22)$ |$0.16(0.2)$ |
> > >
> > > **Policy network: [$\mathrm{dim}(\mathcal{S})$, 512, 512, 512, $\mathrm{dim}(\mathcal{A})$]**
> > > | Algorithm   | Ant-v4 | Humanoid-v4  | HumanoidStandup-v4 |
> > > |--------|-----|------|------|
> > > | PPO   | $5.55(3536.7)$| $30.9(41786.79)$| $48.98(241293.98)$ |
> > > | SPO   | $0.17(0.28)$| $0.17(0.2)$ |$0.17(0.21)$ |
> > >
> > > It can be observed that as the number of network parameters increases, PPO **fails to constrain the probability ratio deviation**, with its maximum value reaching an astonishing **240,000** in the HumanoidStandup-v4 environment. In contrast, regardless of changes in the network structure, SPO's probability ratio deviation remains very stable and stays below the hyperparameter threshold $\epsilon=0.2$. This fully demonstrates that SPO's outstanding performance does not benefit from a specific network architecture but is capable of **consistently constraining the probability ratio**—and thus stabilizing training—under any network settings.
> > >
> > > Finally, thank you for your constructive feedback on our paper. As this is our final response, if all your concerns have been addressed, we would sincerely appreciate your stronger support (i.e., higher rating) for our paper. Thank you very much!
> > >
> > > Best,
> > >
> > > Authors
> > >
> > > ---
> > > *Reference:*
> > >
> > > [1] S Huang et al. The 37 implementation details of proximal policy optimization.

---

### Official Review · Reviewer_8JyT · 2025-03-14

**Overall Recommendation:** 1

**Summary:**

The paper introduces Simple Policy Optimization (SPO), a new unconstrained first-order reinforcement learning algorithm designed to effectively combine strengths from Trust Region Policy Optimization (TRPO) and Proximal Policy Optimization (PPO). SPO modifies PPO's objective by proposing a novel surrogate loss that constrains the probability ratio using Total Variation (TV) divergence, theoretically enhancing policy improvement. Empirical evaluations on Atari and MuJoCo benchmarks suggest SPO can achieve better or comparable performance than PPO, particularly when training deeper neural networks.


### Update After Rebuttal

I appreciate the authors’ response. However, none of my concerns have been adequately addressed.

**Regarding Theorem 4.3 and the associated conclusion:**
It appears that the authors misunderstood the core of my concern. My point is that Theorem 4.3 does *not* support the claim that "TV can lead to better policy improvement." What is actually shown is that the surrogate objective with TV is greater than that with KL, i.e., $L(\pi_{\text{TV}}) \ge L(\pi_{\text{KL}})$. This does not imply that the *true* performance of the final policy is better, i.e., $\eta(\pi_{\text{TV}}) \ge \eta(\pi_{\text{KL}})$. Additionally, Equation (3) should not include a constraint, as the surrogate objective itself does not include one.

**Regarding the rigor of Equations 17–18:**
What I am asking for here is a formal and rigorous proof. I am concerned that the method may be biased due to the added term, and this needs to be clarified with a precise derivation.

**On experimental sufficiency:**
As I previously noted, even adding a random component can result in performance improvements on half of the 57 Atari games. The current experimental evidence is therefore not sufficient to substantiate the paper’s claims.

Given the above concerns, I will maintain my original score. The paper itself may introduce some potential alternative solution to PPO. I hope the author polish their theories and enhance their experiments.

**Claims And Evidence:**

The main claims—that SPO achieves superior policy improvement by effectively bounding probability ratios using TV divergence—are inadequately supported by the presented evidence. Specifically, the theoretical claim that "TV divergence offers a larger solution space than KL divergence" is problematic. The provided proofs confuse constraint-based optimization (eq. 13) and surrogate objective-based optimization (eq. 16). Consequently, the statement that TV constraints yield a "more effective solution space" than KL constraints lacks convincing support and clarity. Additionally, the step from eq. (17) to (18) and the treatment of the absolute advantage versus squared difference are neither adequately justified nor analyzed.

Please also see Theoretical Claims for more detail.

**Essential References Not Discussed:**

Further discussion and comparison with PPO using a KL penalty would be beneficial.

**Experimental Designs Or Analyses:**

The experimental analyses have limitations. The selection of only 35 out of 57 Atari environments raises concerns about potential cherry-picking or hyperparameter sensitivity. The inconsistent results between Figure 1 and Figure 8 suggest sensitivity to hyperparameters, not robustness.


Additionally, the inconsistent performance of PPO-Penalty regarding median metrics versus optimality gaps requires further clarification. Moreover, the method "PPO-Penalty" itself is not clearly defined in the paper; I could not find any explicit definition or description provided.

Finally, deeper network experiments compared only to PPO are insufficient to demonstrate the generality of the proposed method; comparisons to additional baselines are required.

**Methods And Evaluation Criteria:**

In my understanding, the proposed approach differs from PPO with  KL Penalty primarily in two aspects:

- It replaces the KL divergence with alternative divergence terms.
- It introduces an absolute value of the advantage term into the adaptive coefficient.

Therefore, a comparison with this PPO variant would be necessary to clearly demonstrate whether using TV divergence provides any advantage over KL divergence.

**Other Comments Or Suggestions:**

- Formulations can be streamlined for clarity; currently, some equations are unnecessarily repeated or redundant.
- Theorem 3.1 is not actively used and does not aid in understanding the proposed SPO method; it could be omitted or properly integrated.

**Other Strengths And Weaknesses:**

Strengths include the simplicity of the SPO objective and promising empirical results for deeper neural networks. Weaknesses center primarily around theoretical ambiguities, insufficient comparative analysis, and questionable hyperparameter sensitivity.

**Questions For Authors:**

1. Can you rigorously justify why the absolute deviation (eq. 18) is preferred over squared deviation, given significant conceptual differences?
2. How do you explain the contradictory performances of SPO in Figures 1 and 8?
3. What are the detailed reasons for selecting only 35 out of the 57 available Atari environments, and how were these environments chosen?
4. Could you explicitly compare the performance of SPO with PPO-penalty, clearly identifying whether TV divergence or other modifications provide the primary benefit?

**Relation To Broader Scientific Literature:**

The paper appropriately situates itself within the literature of TRPO and PPO improvements. However, th theoretical contributions claimed, especially regarding TV versus KL divergence, must be more clearly contextualized with respect to existing results on divergence constraints in reinforcement learning literature.

**Theoretical Claims:**

Q 1:

The paper contains several clear theoretical and logical errors.

The authors claim that "TV divergence offers a larger solution space compared to methods incorporating a looser KL divergence constraint." However, the fundamental issue is that when optimizing the surrogate objective loss, the paper does not actually enforce a KL divergence constraint. Specifically, in equation (13), restricting the solution space to $\tilde{\pi} \in \Omega_{TV}$ is incorrect. The authors seem to have confused constraint-based optimization objectives with surrogate loss-based objectives.

Consequently, their conclusion that "Optimizing the lower bound with TV divergence constraints offers a more effective solution space than using KL divergence constraints, leading to better policy improvement" is incorrect.

To clarify this rigorously, if the authors wish to demonstrate that surrogate objective $A$ is superior to surrogate objective $B$, they must:

- Remove artificial restrictions on the solution space, such as $\tilde{\pi} \in \Omega_{TV}$.
- Provide a formal proof showing $\eta(\pi^*_A) \geq \eta(\pi^*_B)$ rather than relying solely on comparisons using the surrogate loss $\mathcal{L}$.





Q 2:



The derivation from Eq. (17) to Eq. (18) does not appear rigorous. Specifically, replacing the absolute value $|r - 1|$ with the squared term $(r - 1)^2$ significantly changes the nature of the function. The paper lacks an analysis of the divergence introduced by this transition from Eq. (17) to Eq. (18). Additionally, it is unclear whether this modification will lead to actual policy performance improvements.



Q 3:

Definition 5.1 does not appear clearly motivated or sufficiently rigorous. The notion of an "$\epsilon$-aligned" surrogate is not strong enough to guarantee policy improvement. In fact, many surrogate functions can satisfy the condition of being "$\epsilon$-aligned" yet fail to yield any meaningful performance improvement. Therefore, the practical usefulness and theoretical significance of this definition remain unclear.

---

> ### Author Rebuttal · Authors · 2025-04-01
>
> Dear Reviewer 8JyT,
>
> Thank you for your comment. Below, we will address your concerns.
>
> >The authors claim that "TV divergence offers a larger solution space compared to methods incorporating a looser KL divergence constraint." However, the fundamental issue is that when optimizing the surrogate objective loss, the paper does not actually enforce a KL divergence constraint. Specifically, in equation (13), restricting the solution space to $\tilde{\pi}\in\Omega_ {\mathrm{TV}}$ is incorrect. The authors seem to have confused constraint-based optimization objectives with surrogate loss-based objectives.
>
> Thank you for your comment. However, we respectfully disagree with this perspective. In our approach, we introduce the SPO objective in Equation (16) to constrain the probability ratio, thereby ensuring the $\epsilon$-aligned property defined in Definition 5.1. Furthermore, as established in Equation (9), constraining the probability ratio across batch data inherently bounds the total variation (TV) divergence, which naturally leads to a TV divergence-based trust region. Importantly, Theorem 4.3 demonstrates that the performance improvement lower bound under TV divergence constraints is indeed superior to that under KL divergence constraints.
>
> >The derivation from Eq. (17) to Eq. (18) does not appear rigorous. Specifically, replacing the absolute value $|r-1|$ with the squared term $(r-1)^2$ significantly changes the nature of the function. The paper lacks an analysis of the divergence introduced by this transition from Eq. (17) to Eq. (18). Additionally, it is unclear whether this modification will lead to actual policy performance improvements.
>
> >Definition 5.1 does not appear clearly motivated or sufficiently rigorous. The notion of an "$\epsilon$-aligned" surrogate is not strong enough to guarantee policy improvement. In fact, many surrogate functions can satisfy the condition of being "$\epsilon$-aligned" yet fail to yield any meaningful performance improvement. Therefore, the practical usefulness and theoretical significance of this definition remain unclear.
>
> >Can you rigorously justify why the absolute deviation (eq. 18) is preferred over squared deviation, given significant conceptual differences?
>
> This might be a misunderstanding of our method. We employ the squared penalty because it is convex with respect to the probability ratio $r$, and its optimal solution naturally lies at the probability ratio boundary $r^*=1+\mathrm{sign}(A)\cdot\epsilon$. This demonstrates our method's effectiveness in constraining the probability ratio, which consequently leads to indirect optimization of the policy's performance improvement lower bound (10).
>
> >How do you explain the contradictory performances of SPO in Figures 1 and 8?
>
> Please note that Figure 1 presents results using 50 to 101-layer ResNet architectures, while Figure 8 employs the default shallow CNN structure for comparison.
>
> >What are the detailed reasons for selecting only 35 out of the 57 available Atari environments, and how were these environments chosen?
>
> We deliberately selected a subset of Atari environments where PPO can operate successfully. Among the 57 Atari games, some involve sparse reward settings where neither PPO nor SPO can effectively train reinforcement learning policies. Furthermore, given the substantial computational overhead of Atari experiments, running all environments would be prohibitively time-consuming. We maintain that the environments included in our experiments provide sufficient basis for meaningful algorithmic comparison.
>
> >Could you explicitly compare the performance of SPO with PPO-penalty, clearly identifying whether TV divergence or other modifications provide the primary benefit?
>
> Thank you for your suggestion. Figure 4 in the paper includes comparative experiments between PPO-Penalty and other extensive baselines, with the results clearly demonstrating SPO's performance advantages over both PPO-Clip and PPO-Penalty in MuJoCo environments (10 seeds).
>
> Best,
>
> Authors

---

### Official Review · Reviewer_SqU8 · 2025-03-15

**Overall Recommendation:** 3

**Summary:**

This paper studies an alternative of PPO, named Simple Policy Optimization (SPO), by optimizing a tighter performance lower bound using Total Variation (TV) divergence. The authors are concerned with PPO’s limitation in constraining probability ratios, which is an important problem to study.

**Claims And Evidence:**

Most claims are well supported.

**Essential References Not Discussed:**

The paper [1] also studies replacing the divergence term to TV divergence but is not discussed.

[1] Chu et al. "A Strong On-Policy Competitor To PPO."

**Experimental Designs Or Analyses:**

Yes, the experimental designs are valid. However, it would make the results more convincing and interesting if the following problems were also studied:
1. PPO is widely used in large-scale RL problems, and KL regularization will be incorporated when the policy NN is complex, such as LLM. By doing so, the concerned limitation of PPO in constraining probability ratios would not be a problem and hardly to be observed in practice. Can the authors compare the above PPO variant and SPO? It would be great if the experiment could be completed for more complex NNs, such as LLM alignment or reasoning. The studied 7-layer MLP is still not deep enough to claim SPO's robustness in larger-scale settings.

2. Another interesting ablation will be to decrease PPO's $\epsilon$ or increase SPO's $\epsilon$ so that their average ratio deviation is the same and see if the ratio is the main reason for SPO's better performance.

3. An ablation on SPO's $\epsilon$ will help the readers better understand its robustness.

**Methods And Evaluation Criteria:**

Yes.

**Other Comments Or Suggestions:**

I didn't observe obvious typos. Authors may consider replace "can not" to "cannot".

**Other Strengths And Weaknesses:**

Strengths: 1. The paper is clearly written and easy to follow.\
2. The paper has a good motivation.

Weaknesses: 1. The related works that use TV divergence or more general divergence are not thoroughly discussed.\
2. The experimental results and ablations can be further enhanced. Please see the Experimental Designs Or Analyses part.

**Questions For Authors:**

Can the authors comment on the potential drawbacks of the proposed method or, more generally, using TV divergence, compared to PPO and TRPO, such as instabilities? And will add KL regularization to PPO make it better?

**Relation To Broader Scientific Literature:**

This paper is related to PPO and other zeroth-order policy gradient methods.

**Theoretical Claims:**

I checked the theories that show the properties of the proposed method but did not check the derivation in previous sections.

---

> ### Author Rebuttal · Authors · 2025-04-01
>
> Dear Reviewer SqU8,
>
> Thank you for your constructive feedback. Below, we will address your concerns.
>
> >PPO is widely used in large-scale RL problems, and KL regularization will be incorporated when the policy NN is complex, such as LLM...
>
> We appreciate your suggestion. However, the purpose of KL regularization in PPO for LLMs is to prevent reward over-optimization and catastrophic forgetting caused by policy collapse. This does not prevent PPO's ratio deviation issue, as ratio deviation is defined with respect to the initial policy $\pi_ {\theta_ {\mathrm{old}}}$ at each update. Additionally, due to the limited rebuttal period, we may not be able to provide comparative results of SPO-fine-tuned LLMs. We sincerely apologize for this limitation.
>
> >Another interesting ablation will be to decrease PPO's $\epsilon$ or increase SPO's $\epsilon$ so that their average ratio deviation is the same and see if the ratio is the main reason for SPO's better performance.
>
> >An ablation on SPO's $\epsilon$ will help the readers better understand its robustness.
>
> Thank you for your comment. We have supplemented the following experiments:
>
> | Environment   | Ant-v4 | Humanoid-v4  | HumanoidStandup-v4 |Walker2d-v4 |
> |--------|-----|------|------|------|
> | SPO ($\epsilon=0.1$)   | $5045.05\pm1005.02$| $4866.81\pm1230.83$| $160103.83\pm11509.79$ |$3737.86\pm903.16$ |
> | SPO ($\epsilon=0.2$)   | $4760.69\pm849.75$| $4852.76\pm1414.11$| $178853.55\pm43151.81$ |$2944.91\pm1257.06$ |
> | SPO ($\epsilon=0.3$)   | $4096.41\pm863.53$| $4532.02\pm1319.04$ |$156627.08\pm8519.33$ |$3193.54\pm1148.36$ |
> | SPO ($\epsilon=0.4$)   | $3594.85\pm857.46$| $4555.34\pm1497.48$ |$165393.59\pm29599.54$ |$2423.38\pm1112.68$ |
> | SPO ($\epsilon=0.5$)   | $3258.7\pm854.74$| $2516.6\pm1050.51$ |$149689.79\pm16572.68$ |$2187.09\pm989.88$ |
>
> The results demonstrate that over-large $\epsilon$ (0.5) consistently leads to performance degradation, which aligns with the theoretical lower bound.
>
> >The paper [1] also studies replacing the divergence term to TV divergence but is not discussed.
>
> Thank you for pointing out. We will include this in the related work section in our subsequent revisions.
>
> >The related works that use TV divergence or more general divergence are not thoroughly discussed.
>
> We appreciate the suggestion. However, to our knowledge, [2] is the only existing work that discusses the relationship between total variation (TV) divergence and the policy ratio, we have discussed this paper in our related work section. KL divergence is first proposed in original TRPO paper [3]. To the best of our knowledge, beyond TV divergence and KL divergence, there appear to be no published works employing more general divergence measures in this context.
>
> >I didn't observe obvious typos. Authors may consider replace "can not" to "cannot".
>
> Thank you for your thorough review. We will carefully incorporate your suggestions.
>
> >Can the authors comment on the potential drawbacks of the proposed method or, more generally, using TV divergence, compared to PPO and TRPO, such as instabilities? And will add KL regularization to PPO make it better?
>
> In fact, we compared a PPO variant with KL regularization (PPO-Penalty), as shown in Figure 4 of our paper. The results demonstrate that SPO achieves superior performance.
>
> Best,
>
> Authors
>
> ---
> *Reference:*
>
> [1] X Chu et al. A strong on-policy competitor to PPO.
>
> [2] J Queeney et al. Generalized proximal policy optimization with sample reuse.
>
> [3] J Schulman et al. Trust region policy optimization.

---

> > ### Comment · Reviewer_SqU8 · 2025-04-02
> >
> > Thank the authors for the response. The rebuttal addressed some of my concerns. However, it is still not clear why PPO+KL does not address the ratio deviation issue, since KL regularization is also added between $\pi$ and the initial policy $\pi_ {\theta_ {\mathrm{old}}}$, similar to how ratio deviation is defined. Reviewer 8JyT also shares similar concerns, but I didn't find satisfying answers in both responses. Besides, the authors said they would compare with [1] in the revision, but didn't state the connections and differences. Other parts of my concerns are addressed.

---

> > > ### Author Response · Authors · 2025-04-03
> > >
> > > Dear Reviewer SqU8,
> > >
> > > Thank you for your additional comments. To address your concerns, as per your request, we have supplemented the performance of PPO+KL, POP3D, and SPO in the MuJoCo environment:
> > >
> > > | Algorithm   | Ant-v4 | Humanoid-v4  | HumanoidStansup-v4 | Walker2d-v4 |
> > > |--------|-----|------|------|------|
> > > | PPO+KL    | $1588.32\pm663.4$| $764.03\pm108.08$| $135953.03\pm26110.9$ | $2563.23\pm912.8$ |
> > > | POP3D   | $-490.43\pm674.98$| $349.93\pm158.44$ |$70518.6\pm20164.79$ | $569.14\pm120.94$ |
> > > | SPO   | $3900.77\pm1099.6$| $3771.03\pm1461.88$ |$159541.83\pm35398.75$ | $2416.72\pm804.08$ |
> > >
> > > For PPO+KL, we employed an adaptive penalty coefficient beta based on the KL divergence. For POP3D, we referred to the hyperparameter settings in [1]. The above results were obtained using a 7-layer network across 4 random seeds. As observed, SPO still achieved the best performance.
> > >
> > > Additionally, we found that as the network deepens, POP3D often fails to effectively constrain the probability ratio and KL divergence, leading to suboptimal performance. Moreover, naively applying PPO with KL divergence penalty does not appear to effectively enhance its performance.
> > >
> > > Regarding the difference between SPO and POP3D in [1], we believe the most fundamental distinction lies in the fact that SPO imposes a specific penalty coefficient $\frac{|\hat{A}(s_t,a_t)|}{2\epsilon}$ on each data pair $(s_t,a_t)$, whereas POP3D defines a **point probability distance** as a lower bound for the TV divergence.
> > >
> > > However, in practice, we can achieve a similar effect by directly constraining the probability ratio, because the probability ratio deviation $|\frac{\tilde{\pi}(a_t|s_t)}{\pi(a_t|s_t)}-1|$ with a sufficiently large batch size approximately equals the TV divergence $D_ {\mathrm{TV}}(\pi\Vert\tilde{\pi})$.
> > >
> > > Limiting the probability ratio deviation $|\frac{\tilde{\pi}(a_t|s_t)}{\pi(a_t|s_t)}-1|$ is more straightforward because the probability ratio also appears in the **surrogate objective** $\frac{\tilde{\pi}(a_t|s_t)}{\pi(a_t|s_t)}\cdot\hat{A}(s_t,a_t)$, which naturally leads to
> > >
> > > $\max_ {\tilde{\pi}}\frac{\tilde{\pi}(a_t|s_t)}{\pi(a_t|s_t)}\cdot\hat{A}(s_t,a_t)$
> > >
> > > $\mathrm{s.t.}|\frac{\tilde{\pi}(a_t|s_t)}{\pi(a_t|s_t)}-1|\leq\epsilon\Leftrightarrow 1-\epsilon\leq\frac{\tilde{\pi}(a_t|s_t)}{\pi(a_t|s_t)}\leq 1+\epsilon$
> > >
> > > Using a penalty function, and to ensure convexity and differentiability, a squared penalty can be employed. As a result, we are trying to optimize
> > >
> > > $J(\theta)=\frac{\tilde{\pi}(a_t|s_t)}{\pi(a_t|s_t)}\cdot\hat{A}(s_t,a_t)-k\cdot(\frac{\tilde{\pi}(a_t|s_t)}{\pi(a_t|s_t)}-1)^2=r_t(\theta)\cdot\hat{A}(s_t,a_t)-k\cdot(r_t(\theta)-1)^2$
> > >
> > > Based on the previous analysis, we want the probability ratio deviation $|r_t(\theta)-1|$ to be constrained because its expectation is the TV divergence. Therefore, the boundary for the probability ratio is $1+\mathrm{sign}(\hat{A}(s_t,a_t))\cdot\epsilon$.
> > >
> > > Note that $J(\theta)$ is a quadratic function of $r_t(\theta)$. Therefore, we want its extremum to lie exactly on the constraint boundary $1+\mathrm{sign}(\hat{A}(s_t,a_t))\cdot\epsilon$, ensuring that the probability ratio deviation $|r_t(\theta)-1|$ will be constrained as long as the number of iterations is sufficient. Then we will find that when $k=\frac{|\hat{A}(s_t,a_t)|}{2\epsilon}$, this condition is exactly satisfied, which ultimately leads to the objective of SPO:
> > >
> > > $J(\theta)=\frac{\tilde{\pi}(a_t|s_t)}{\pi(a_t|s_t)}\cdot\hat{A}(s_t,a_t)-\frac{|\hat{A}(s_t,a_t)|}{2\epsilon}\cdot(\frac{\tilde{\pi}(a_t|s_t)}{\pi(a_t|s_t)}-1)^2=r_t(\theta)\cdot\hat{A}(s_t,a_t)-\frac{|\hat{A}(s_t,a_t)|}{2\epsilon}\cdot(r_t(\theta)-1)^2.$
> > >
> > > For POP3D, its core idea is similar to that of SPO, as both aim to constrain the TV divergence. However, SPO ensures stable probability ratio deviation due to its adaptive penalty coefficients $k=\frac{|\hat{A}(s_t,a_t)|}{2\epsilon}$ and convexity.
> > >
> > > We sincerely appreciate your valuable feedback on our paper. As this is our final response, if all your concerns have been addressed, we would also kindly ask for your stronger support for our paper. Thank you!
> > >
> > > Best,
> > >
> > > Authors
> > >
> > > ---
> > > *Reference:*
> > >
> > > [1] X Chu et al. A strong on-policy competitor to PPO.

---

### Decision · Program_Chairs · 2025-05-01

**Decision:**

Accept (poster)

**Comment:**

This paper propose a new policy gradient algorithm SPO.

The strength includes

(1) simplicity: SPO only requires single-line modification to the code of PPO algorithm.
(2) well-justified theoretical motivation: the paper points out a limitation of PPO in that it can fail to bound the probability ratios between successive policies despite trying to do so. And the proposed SPO update rule solves this issue.
(3) the claim is also backed up by extensive experimental evaluation, including the Atari and Mujuco benchmarks with various neural network architecture.

There is concern raised among the reviewers regarding the theory-practice mismatch. The analysis provided in the paper is on the TV constrained version of the optimization objective while the actual algorithm takes the Lagrangian form. While this is indeed not ideal, it's arguably quite common in the RL literature, such as in the original TRPO and PPO paper.

Overall, despite that, I believe this is a solid contribution to the suite of scalable RL algorithms.